# Washable Antimicrobial Wipes Fabricated from a Blend of Nanocomposite Raw Cotton Fiber

**DOI:** 10.3390/molecules28031051

**Published:** 2023-01-20

**Authors:** Sunghyun Nam, Doug J. Hinchliffe, Matthew B. Hillyer, Lawson Gary, Zhongqi He

**Affiliations:** 1U.S. Department of Agriculture, Agricultural Research Service, Southern Regional Research Center, New Orleans, LA 70124, USA; 2Wildwood Cotton Technologies, Greenwood, MS 38930, USA

**Keywords:** raw cotton, silver nanoparticle, nanocomposite, antimicrobial property, wipes, wash-durable

## Abstract

In this study, a simple and effective way to produce washable antimicrobial wipes was developed based on the unique ability of raw cotton fiber to produce silver nanoparticles. A nanocomposite substructure of silver nanoparticles (25 ± 3 nm) was generated in raw cotton fiber without reducing and stabilizing agents. This nanocomposite raw cotton fiber (2100 ± 58 mg/kg in the concentration of silver) was blended in the fabrication of nonwoven wipes. Blending small amounts in the wipes—0.5% for antimicrobial properties and 1% for wipe efficacy—reduced the viability of *S. aureus* and *P. aeruginosa* by 99.9%. The wipes, fabricated from a blend of 2% nanocomposite raw cotton fiber, maintained their antibacterial activities after 30 simulated laundering cycles. The washed wipes exhibited bacterial reductions greater than 98% for both Gram-positive and Gram-negative bacteria.

## 1. Introduction

The recent global pandemic outbreak has created awareness about the importance of personal protection, resulting in a remarkable increase in the use of antimicrobial wipes. People have been sanitizing or disinfecting hard surfaces in their homes, including countertops, bathroom surfaces, desktops, and doorknobs/handles, more frequently. Moreover, the usage of wipes in the medical sector, hotels, restaurants, and schools is increasing. With a compound annual growth rate of 11%, the worldwide antimicrobial wipes market is predicted to reach $21.6 billion in 2030 [1]. Most of these wipes are discarded after a single use, generating remarkable amounts of waste. Approximately, 3.8 million tons of wipes are thrown into landfills each year [2]. The wipe waste can stay in landfills for hundreds of years and become a source of environmental microplastic fiber pollution.

To reduce the number of discarded wipes, we have developed washable antimicrobial wipes made from a blend of nanocomposite raw cotton fiber. Although nanocomposites offer advantages in creating functionalities, embedding nanoparticles into the cotton fiber, while preserving its naturally occurring internal structure, is challenging. Cotton fiber is a highly crystalline fiber and has a complex hierarchical structure. A study that tried to mix magnesium hydroxide nanoparticles into cellulose dissolved in ionic liquid reported a significant aggregation of nanoparticles [3]. This result indicates that cotton cellulose is not readily compatible with these nanoparticles. The lack of feasible methods for filling nanoparticles—with no aggregation—into cotton fiber has led to little success in the production of nanocomposite cotton fibers.

We have been studying the in situ synthesis of silver nanoparticles inside the cotton fiber and successfully obtained the internal dispersion of the nanoparticles [4,5,6]. Silver nanoparticles are a popular antimicrobial agent used for producing odor-inhibiting and anti-infective textile products [7]. Our method includes the formation of silver complex ions, i.e., [Ag(NH_3_)_2_]^+^, in the alkali-swollen cotton fiber followed by the introduction of a reductant aqueous fluid. Without using any stabilizing agents—which are generally needed to prevent the aggregation of nanoparticles—silver nanoparticles were generated within the fiber. Another method is to utilize the condensed tannins present in naturally colored brown cotton fiber as a reducing agent, producing a nanocomposite brown cotton fiber [6]. Condensed tannins (proanthocyanidins), which are responsible for the brown color [8,9,10], contain phenolic groups. These groups participate in redox reactions by donating electrons to silver ions and forming quinones.

In this study, we produced washable antimicrobial wipes by introducing a nanocomposite structure into the raw cotton fiber. Raw cotton fiber contains pectin, sugars, fatty alcohols, and hemicellulose. These non-cellulosic components were used as reducing agents, producing silver nanoparticles within the raw cotton fiber. The obtained nanocomposite raw cotton fiber was blended in the fabrication of nonwoven wipes with different weight ratios, and their antimicrobial performance and wipe efficacy were assessed. The wash durability of the nanoparticles embedded in the raw cotton fiber was evaluated by conducting simulated home launderings for 30 cycles.

## 2. Results

The ability of raw cotton fiber to produce silver nanoparticles was confirmed by comparing it with that of scoured/bleached cotton fiber. Scouring and bleaching are typical pretreatments conducted prior to finishing or dyeing in the textile industry. Scouring (i.e., boiling the fiber in hot, dilute, aqueous sodium hydroxide [11]) removes most non-cellulosic constituents (such as wax, pectin, and proteins) from cotton fiber. Bleaching removes the natural pigment present in raw cotton fiber. These processes consume a plethora of chemicals, energy, and water. For example, hydrogen peroxide bleaching requires 0.118 kg/kg textiles of chemical, 8.34 MJ/kg textiles of energy, and 50 L/kg textiles of water [12]. A simple one-step process of heat treatment in an aqueous solution of silver precursor (AgNO_3_) can activate the synthetic reaction in the raw cotton. Figure 1 shows the photographs of raw and scoured/bleached cotton fibers before, during, and after the heat treatment (from top to bottom). Untreated raw cotton fibers are slightly yellow because of a trace amount of natural pigment [11]. The raw cotton fiber turned yellow within 5 min of the heat treatment and turned brown after 30 min. However, the scoured/bleached cotton fiber exhibited a negligible change in color after 30 min. This unique optical property of the treated raw cotton fiber results from the surface plasmon resonance of the nanoparticles. Surface plasmon resonance occurs when conduction electrons of metallic nanoparticles oscillate in response to the alternating electric field of incident electromagnetic radiation [13]. Silver nanoparticles show a strong surface plasmon resonance at wavelengths of 410–420 nm [14]. This spectral feature depends on various factors including the shape and size of the particle and its surrounding medium. In the cotton medium, the surface plasmon resonance of sphere-like silver nanoparticles with about 25 nm in diameter was observed at 420 nm [15].

To confirm the internal synthesis of the silver nanoparticles, the cross-section of the treated raw cotton fiber was observed using TEM. Figure 2a shows that numerous nanoparticles were generated and embedded in the primary wall of the raw cotton fiber. The observed primary wall was approximately 700 nm thick. The primary cell wall, which is one of the substructures of cotton fiber, contains non-cellulosic constituents. Some of the non-cellulosic constituents, such as pectin and sugars, have reducing properties. Silver positive ions (Ag^+^) were diffused into the swollen raw cotton fiber in an aqueous solution and were reduced into silver atoms (Ag^0^) through the oxidation of non-cellulosic constituents. The silver atoms nucleate in small clusters, which grow into particles. The nanoparticles were formed underneath the fiber surface.

The TEM image taken at higher magnification (Figure 2b) shows that the nanoparticles are sphere-like and are dispersed without aggregation. Nanoparticles tend to aggregate into larger particles. To prevent particle aggregation, stabilizing agents are typically used during the synthesis of nanoparticles. In the in situ synthesis within the raw cotton fiber, no stabilizing agents were needed. Silver cations bind to the oxygen-rich components of the cotton fiber, allowing particles to grow on the solid surface of the structural elements (i.e., microfibrils). This surface-pinned synthesis prevents particle aggregation. The high-resolution TEM image of a single particle (Figure 2c) shows that lattice fringes are joined at different angles, indicating that particles grew through the coalescence of smaller crystallites. The d-spacing determined from the defined lattice fringes was 0.267 nm. The selected area electron diffraction (SAED) pattern (Figure 2d) shows the four concentric rings corresponding to the (1 1 1), (2 0 0), (2 2 0), and (3 1 1) planes of the face-centered cubic structure of elemental silver.

The silver nanoparticles produced inside raw cotton fiber were 25 ± 3 nm in diameter, and their concentration, based on the dry weight of the raw cotton fiber, was 2100 ± 58 mg/kg. Using this nanocomposite raw cotton fiber, nonwoven wipes were fabricated in the nonwoven pilot plant at the Southern Regional Research Center (New Orleans, LA, USA). Figure 3 shows selected photographs of the nonwoven manufacturing process.

The nanocomposite raw cotton fiber was blended with pristine scoured/bleached cotton fibers at various weight ratios on a laboratory card (John D. Hollingsworth on Wheels, Inc., Greenville, SC, USA), producing a fiber web (Figure 3a). The card web was fed into a needle-punch machine (Technoplants srl., Pistoia, Italy) equipped with boards of three barb needles (Groz-Beckert KG, Albstadt, Germany) (Figure 3b). The needle-punching process was conducted to increase the fabric’s strength to withstand multiple washes. The needling process was conducted at 490 strokes/min with a production rate of 5 m/min. The resulting needle-punched fabrics were converted to hydroentangled fabrics on a Fleissner hydroentanglement system (Trützschler Nonwovens GmbH, Dülmen, Germany), in which each strip on the pressure heads consists of 16 orifices/cm with an orifice pore size of 120 µm (Figure 3c). The hydroentangled fabrics were produced at a rate of 5 m/min using a water jet pressure of 2.5 and 9 MPa for prewetting and fiber bonding, respectively. After the hydroentanglement, the fabrics were transferred to the drying oven (Trützschler Nonwovens GmbH) at 170 °C and were wound into rolls (Figure 3d). The area density of the obtained nonwoven wipes was 97 ± 5 g/m^2^ (measured according to ASTM D 6242-98: Test Method for Mass Unit Area of Nonwoven Fabrics).

Figure 4 shows the ultraviolet/visible (UV/vis) reflectance spectra of nonwoven fabrics containing various percentages of the nanocomposite raw cotton fiber (0–20%). The reflectance of the control fabric (containing no nanocomposite raw cotton fiber) was above 80% within a wavelength range of 400–1200 nm. The formation of silver nanoparticles caused a steep drop in the reflectance of the fabric between 400–500 nm, developing a surface plasmon resonance peak. With the increase in the content of nanocomposite raw cotton fiber, the surface plasmon resonance peak became well-defined and gradually strengthened. To examine the effect of silver nanoparticles on the color of the wipes, the diffuse reflectance spectra were obtained within 380–780 nm, from which the CIE LAB color coordinates were calculated. Figure 5 shows the CIE LAB color coordinates including *L**, *a**, and *b** values of the nonwoven fabric as a function of the percentage of nanocomposite raw cotton fiber. *L** represents the lightness of the color. For example, *L** = 0 indicates black and *L** = 100 indicates white. *a** represents the relative position between red and green. A negative value of *a** indicates green, and a positive value of *a** indicates red. *b** represents the relative position between yellow and blue. A negative value of *b** indicates blue, and a positive value of *b** indicates yellow. With the increase in the percentage of the nanocomposite raw cotton fiber, *L** decreased, whereas *a** and *b** both increased. A linear increase was observed in the coordinates of *a** and *b** as a function of the percentage of the nanocomposite raw cotton fiber. The ∆*E** was within 5% for the 1% blend and reached approximately 30% for the 20% blend (Figure 5c).

The antibacterial properties of the wipes fabricated with 0.2%, 0.5%, 1%, 2%, and 5% nanocomposite raw cotton fibers were tested. Figure 6 shows the colony-forming unit counts of *S. aureus* and *P. aeruginosa*, respectively. The corresponding percent viability and log reductions are presented in Table 1. The pristine scoured/bleached cotton wipes (0% blend) exhibited no reductions against *S. aureus* and *P. aeruginosa*. Adding a very small amount (0.5%) of the nanocomposite raw cotton fiber to the fabric remarkably increased the antibacterial activities of the wipes to >99% reductions against both Gram-positive and Gram-negative bacteria. This result indicates the highly potent biocidal activities of silver nanoparticles. The biocidal activity of the silver nanoparticles is primarily attributed to their acting as a source of silver ions. Silver nanoparticles release silver ions through desorption from the surface of particles or by oxidative dissolution [16]. Silver ions are known to deactivate cellular enzymes [17,18] and disable the replication ability of DNA [19]. The large surface area of the silver nanoparticles promotes the release of silver ions.

The colony-forming unit counts of *S. aureus* and *P. aeruginosa* were measured using the wipe efficacy test. The procedure of wiping the inoculated slide involves a total of six passes, i.e., over, back, over, back, over, and back. Figure 7 shows the colony-forming unit counts as a function of the percentages of nanocomposite raw cotton fibers. In the case of *S. aureus*, some viability reduction (41.2%) was observed, suggesting that the wiping procedure alone is moderately effective in killing Gram-positive bacteria. No reduction in *P. aeruginosa* occurred. Table 2 shows the calculated percentages of viability reductions and log reductions. Wiping with nonwoven fabrics containing even 1% of the nanocomposite raw cotton fiber reduced >99% of the *S. aureus* and 99.99% of the *P. aeruginosa*.

To examine the durability of the silver nanoparticles formed underneath the fiber surface, consecutive simulated home launderings were conducted. Figure 8a shows the percentage change in the surface plasmon resonance intensity (*I*_SPR_) of wipes fabricated with 1% and 2% nanocomposite raw cotton fibers as a function of the number of laundering cycles. The loss percentages in *I*_SPR_ after five laundering cycles were approximately 17% and 26% for the 1% and 2% blend wipes, respectively. The dependence of the loss in *I*_SPR_ on the number of laundering cycles decreased with the increase in the number of laundering cycles. The *I*_SPR_ losses of the 1% blend and 2% blend wipes after 30 laundering cycles were approximately 26% and 36%, respectively. Figure 8b shows the amount of silver measured using inductively coupled plasma-mass spectrometry (ICP-MS) based on the dry weight of the wipe as a function of the number of laundering cycles. A similar nonlinear decreasing pattern, i.e., a larger initial drop after the first 5 cycles and a relatively smaller drop after 30 cycles, was observed for the silver concentration. Approximately 24% and 31% of the total silver introduced remained in the 1% and 2% blend wipes, respectively, after 30 laundering cycles.

Figure 9a,b show the percentage reductions in *S. aureus* and *P. aeruginosa*, respectively, for the wipes containing 1% and 2% nanocomposite raw cotton fibers before and after 30 laundering cycles. The wipes made from the 1% blend lost their viability reductions, which were approximately 28% and 35% for *S. aureus* and *P. aeruginosa*, respectively, after 30 laundering cycles. The wipes made from the 2% blend exhibited reductions of >98% of both Gram-positive and Gram-negative bacteria after 30 launderings. This result suggests that blending a small amount of nanocomposite raw cotton fiber is a simple and effective way to produce washable antimicrobial wipes.

## 3. Discussion

The increasing use of disposable wipes poses risks to the environment. To reduce the amount of wipe waste discarded into landfills, washable antimicrobial wipes were developed using nanocomposite raw cotton fiber. The antimicrobial silver nanoparticles (25 ± 3 nm in diameter) were produced in situ within the substructure of raw cotton fiber (2100 ± 58 mg/kg in concentration) without any external agents. Simply blending a small amount (1%) of the nanocomposite raw cotton fibers in the nonwoven fabrication imparted wipe efficacy, killing >99% Gram-positive and Gram-negative bacteria. The 2% blend maintained antimicrobial properties (>99%) that were persistent after 30 laundering cycles. This approach is economical and ecofriendly in production because it eliminates the use of reducing and stabilizing agents, which are necessary for the typical chemical synthesis of nanoparticles as well as it uses raw (unscoured and unbleached) cotton fibers. Scouring and bleaching processes not only consume significant amounts of chemicals along with water and energy but also cause high chemical and biological oxygen demands in textile effluents [20]. The use of nanocomposite raw cotton fibers is not limited to nonwoven fabrication but is applicable to the fabrication of yarns for durable antimicrobial woven products, such as socks, underwear, and sportswear.

## 4. Materials and Methods

### 4.1. Materials

The mechanically cleaned raw cotton fiber was provided by the industrial collaborative partner (Wildwood Cotton Technologies, Greenwood, MS, USA). Scoured and bleached cotton fiber was obtained through a donation from Barnhardt Manufacturing Co. (Charlotte, NC, USA). Silver nitrate (AgNO_3_, 99.9%) was purchased from J. T. Baker (Radnor, PA, USA). Other chemicals including Triton X-100, methyl methacrylate, butyl methacrylate, and methyl ethyl ketone were reagent-grade and purchased from Sigma-Aldrich (St. Louis, MO, USA). They were used as received without further purification. In-house prepared deionized (DI) water was used as a solvent for preparatory reaction experiments and tap water was used for wash durability testing.

### 4.2. Nanocomposite Raw Cotton Fiber Preparation

The production of the nanocomposite raw cotton fibers was adapted from a procedure reported previously [15]. Briefly, approximately 5 g of raw cotton fibers was immersed in a 50 mL aqueous solution of AgNO_3_ (5 mM) and Triton X-100 (0.05 wt.%). The immersed sample (i.e., fully wetted cotton fibers) was incubated at 100 °C for 30 min. At the end of the reaction, the fibers were transferred into DI water and washed at least five times, and air-dried.

### 4.3. Structural and Chemical Characterization

The illustrative photos of the nonwoven manufacturing process and macrostructural images of the samples were taken using a digital camera (RX100, Sony, Tokyo, Japan). The cross-section of the fiber samples was examined by a TEM (JEM-2010, Jeol, Tokyo, Japan) and SAED operating at 200 kV. To prepare the cross-sectioned fibers, a bundle of fibers, randomly collected, was combed and embedded in a mixture of methyl methacrylate and butyl methacrylate. Following a published technique [21,22], a UV cross-linker (UVP, CL-1000) was applied for 30 min to polymerize the embedding medium. Then, a block of the fiber bundle was cut into approximately 100 nm-thick slices using a PowerTome Ultramicrotome (Boeckeler Instruments, Inc., Tucson, AZ, USA). After the slices were placed on a carbon-film-coated copper grid, the embedding medium was removed using methyl ethyl ketone. Image J software (version 1.53e) [23] was used to determine the size of nanoparticles.

The concentration of silver in nanocomposite raw cotton fibers was measured using an inductively coupled plasma-mass spectrometer (ICP-MS) through the analytical service provided by the ICP-MS Metals Laboratory at the University of Utah (Salt Lake City, UT, USA). The analysis solution (digests) was prepared by treatment of approximately 0.05 g of the sample with 2 mL of 16 M nitric acid (trace metal grade) and digested in a Milestone Ethos microwave system. These digests were then diluted by weight 1:10, and 10 ppb of indium as an internal standard was added. The digested solution was analyzed with an external calibration curve obtained using a silver single element standard (Inorganic Ventures, Christiansburg, VA, USA).

UV/vis spectra of solid cotton samples were collected using a UV/vis/NIR spectrometer (ISR-2600, Shimadzu, Kyoto, Japan) equipped with an integrating sphere unit. Reflectance spectra were collected in a wavelength range of 220–1200 nm. Color profiles of these samples were evaluated by diffuse reflectance spectra recorded in a wavelength range of 380–780 nm. The obtained spectral data were used to derive the CIE LAB color coordinates (*L**, *a**, and *b**) using UV-2401PC Color Analysis software. The color difference (Δ*E**) from the reference sample (control cotton) was calculated using the following equation:(1)$$\Delta E^{*}=\sqrt{{\left(L^{*}-L_{r}^{*}\right)}^{2}+{\left(a^{*}-a_{r}^{*}\right)}^{2}+{\left(b^{*}-b_{r}^{*}\right)}^{2}}$$
where *L**, *a**, and *b** are the color coordinates for the tested sample, and *L*_r_*, *a*_r_*, and *b*_r_* are the coordinates for the reference sample. The average value of five measurements is presented.

The examination of the antibacterial activity of samples was conducted using the AATCC test method 100–2004: “Antibacterial Finishes on Textile Materials: Assessment of”. Gram-positive *Staphylococcus aureus* (*S. aureus*) ATCC 6538 and Gram-negative *Pseudomonas aeruginosa* (*P. aeruginosa*) ATCC 9027 were used, and the testing was conducted in Microchem Laboratory (Round Rock, TX, USA). The antibacterial activity was evaluated by measuring the percentage reduction in bacteria using the following equation:(2)$$Percent\;reduction\;\left(\%\right)=\left(\frac{B-A}{B}\right)\times 100$$
where *A* is the number of viable test bacteria on the test sample after 24 h of contact time, and *B* is the number of viable test bacteria on the control sample immediately after inoculation.

The wipe efficacy of samples was evaluated in Microchem Laboratory (Round Rock, TX, USA) using the modified ASTM International Method E1153. This method is a quantitative test method designed to evaluate the antimicrobial efficacy of sanitizers on pre-cleaned inanimate, nonporous, non-food contact surfaces. The test microorganisms were Gram-positive *Staphylococcus aureus* (*S. aureus*) ATCC 6538 and Gram-negative *Pseudomonas aeruginosa* (*P. aeruginosa*) ATCC 9027. The wiping procedure was six total passes per inoculated slide including over, back, over, back, over, and back. The procedure for folding the wipe is to fold the wipe in half lengthwise twice and roll the wipe up five times. The area of the wipe was rotated to expose a maximum amount of the wipe surface during the course of the wiping procedure. The contact was 10 min. At the conclusion of the contact time, test carriers are chemically neutralized. The inoculation concentrations for *P. aeruginosa* and *S. aureus* were 6.5 × 10^5^ CFU/mL and 1.42 × 10^6^ CFU/mL, respectively. The neutralizer used in this study was 20 mL of Dey/Engley Broth additionally supplemented to contain 0.1% sodium thiosulfate. Test carriers were enumerated using the standard pour-plating method, resulting in colonies within the agar as well as on the surface. The percentage bacterial reductions were obtained using Equation (2).

The durability test was conducted in house following the AATCC Test Method 61-2007: Colorfastness to Laundering: Accelerated. Using the laboratory machine (Launder-Ometer, M228-AA, SDL Atlas LLC, Rock Hill, SC, USA), 200 mL of detergent solution (AATCC standard reference detergent 124, 0.37 wt.% in tap water) was added into a stainless-steel canister along with ten stainless steel balls (6.35 mm in diameter) to simulate friction during laundering and was preheated to 45 ± 1 °C. In each run, a rectangular nonwoven fabric specimen (50 mm × 100 mm) was tested. After being placed into the preheated canister, the fabric was subjected to rotation at a constant temperature of 40 ± 1 °C with a constant rate of 40 ± 2 rpm for 45 min. After washing, the specimen was rinsed in tap water at room temperature for 5 min and then air-dried for further analyses.

## Figures and Tables

**Figure 1 molecules-28-01051-f001:**
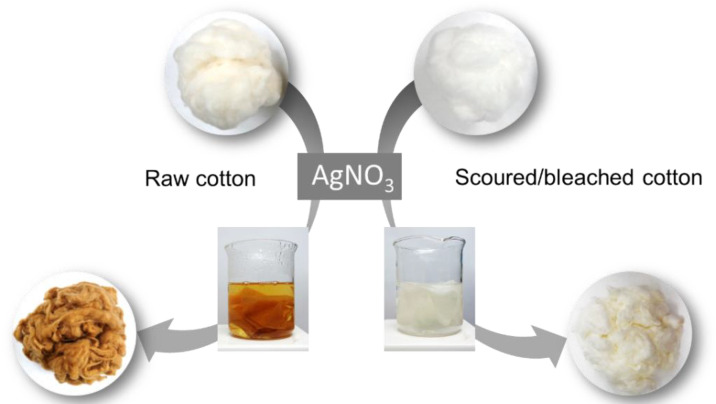
Photographs of raw and scoured/bleached cotton fibers before, during, and after the heat treatment in an aqueous solution of AgNO_3_ (from top to bottom). The color change of the fibers resulting from the surface plasmon resonance of silver nanoparticles indicates that raw cotton fiber can effectively produce silver nanoparticles, but scoured/bleached cotton fiber cannot.

**Figure 2 molecules-28-01051-f002:**
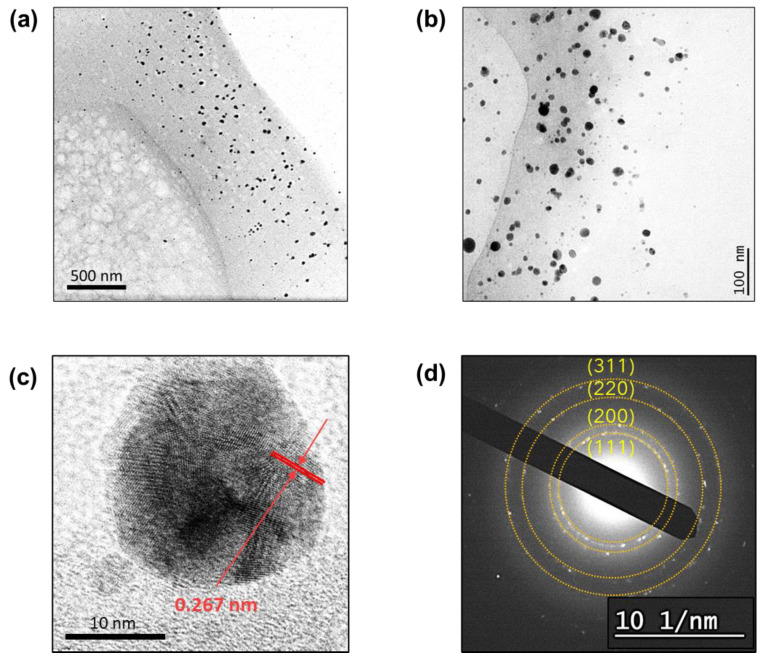
TEM images of the cross-sections of the raw cotton fiber after the in situ synthesis of silver nanoparticles at (**a**) low and (**b**) high magnifications. (**c**) High-resolution TEM image of a single silver nanoparticle and (**d**) SAED pattern taken on the cross-sectioned fiber.

**Figure 3 molecules-28-01051-f003:**
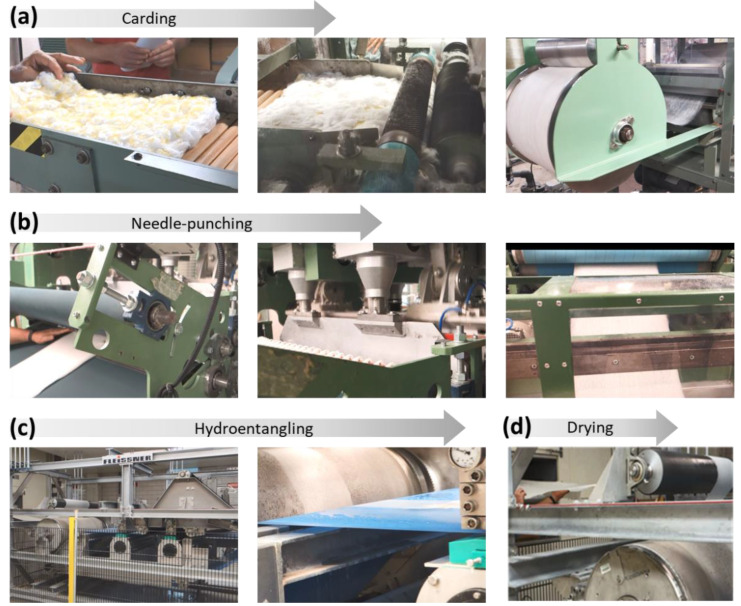
Photographs of the nonwoven manufacturing process of washable antimicrobial wipes containing nanocomposite raw cotton fiber: (**a**) blending and carding, (**b**) needle-punching, (**c**) hydroentanglement, and (**d**) drying.

**Figure 4 molecules-28-01051-f004:**
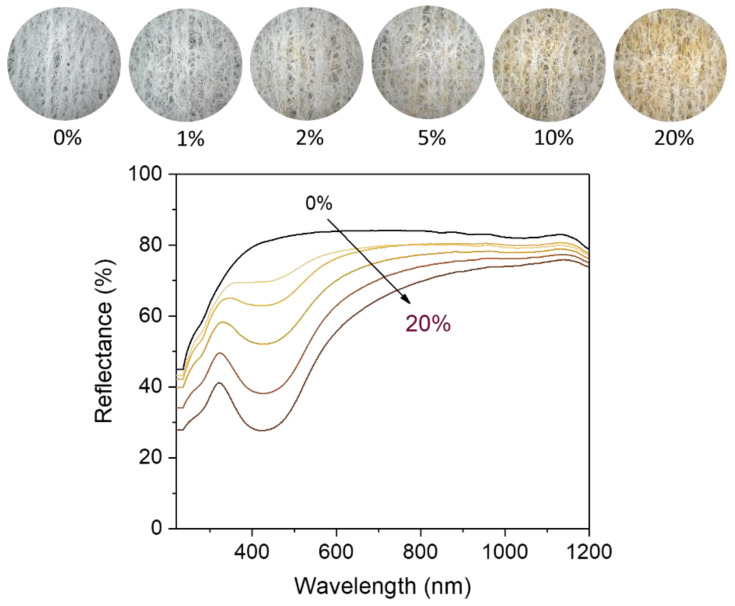
Photographs (**top**) and UV/vis reflectance spectra (**bottom**) of nonwoven wipes fabricated from nanocomposite raw cotton fibers with various weight percentages.

**Figure 5 molecules-28-01051-f005:**
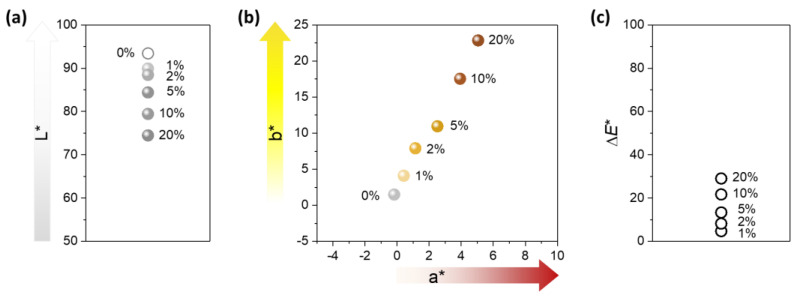
CIE LAB color coordinates-(**a**) *L** and (**b**) *a** and *b**-and (**c**) color difference (Δ*E**) of nonwoven wipes fabricated from nanocomposite raw cotton fibers with various weight percentages.

**Figure 6 molecules-28-01051-f006:**
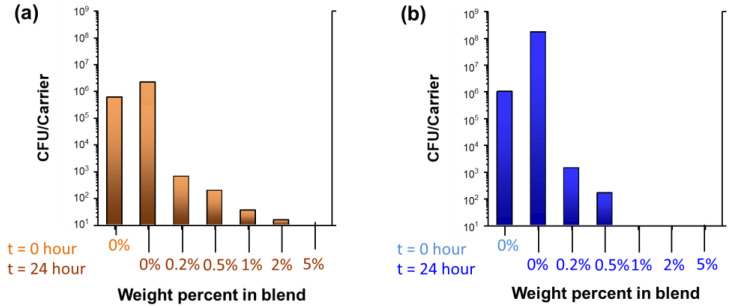
Colony-forming unit counts (CFU) of (**a**) *S. aureus* and (**b**) *P. aeruginosa* for nonwoven wipes fabricated from nanocomposite raw cotton fibers with various weight percentages.

**Figure 7 molecules-28-01051-f007:**
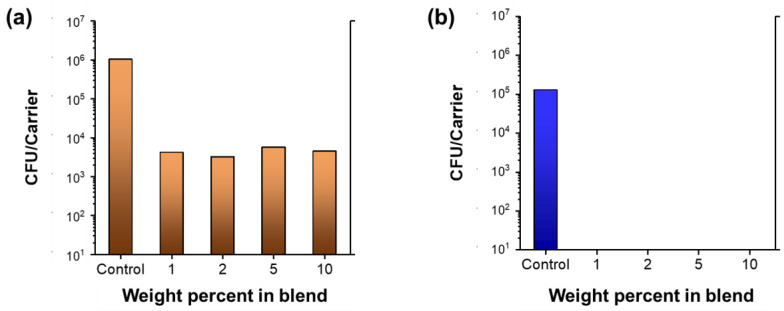
Colony-forming unit counts (CFU) of (**a**) *S. aureus* and (**b**) *P. aeruginosa* obtained from the wipe efficacy test for nonwoven fabrics fabricated from nanocomposite raw cotton fibers with various weight percentages.

**Figure 8 molecules-28-01051-f008:**
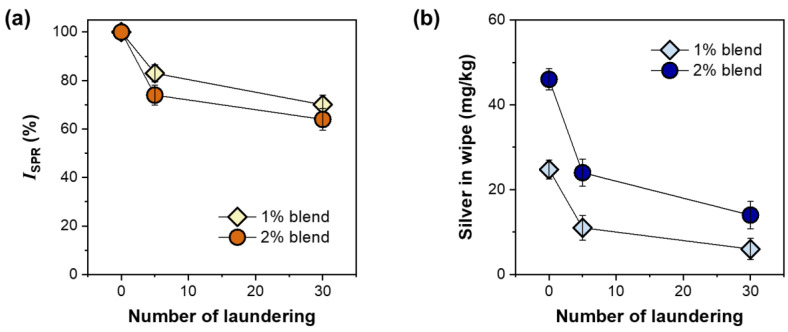
(**a**) Intensity of surface plasmon resonance peak (*I*_SPR_) and (**b**) the concentration of silver for nonwoven wipes fabricated from 1% and 2% blends of nanocomposite raw cotton fibers.

**Figure 9 molecules-28-01051-f009:**
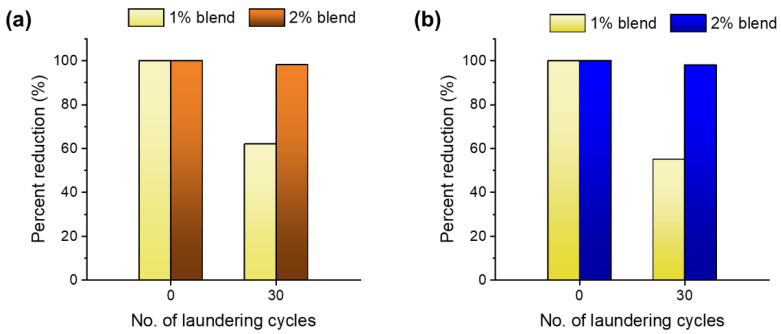
Percentage viability reductions in (**a**) *S. aureus* and (**b**) *P. aeruginosa* for nonwoven wipes fabricated from blends of 1% and 2% nanocomposite raw cotton fibers before and after 30 laundering cycles.

**Table 1 molecules-28-01051-t001:** Percentage reductions and log10 reductions in *S. aureus* and *P. aeruginosa* for nonwoven wipes fabricated from nanocomposite raw cotton fiber with various weight percentages.

Weight Percent in a Blend		0%	0.5%	1%	2%	5%	10%
*S. aureus*	Percent reduction	0	99.97%	99.99%	99.99%	99.99%	99.99%
	Log_10_ reduction	0	3.52	4.77	4.77	4.77	4.77
*P. aeruginosa*	Percent reduction	0	99.92%	99.99%	99.99%	99.99%	99.99%
	Log_10_ reduction	0	3.08	5.00	5.00	5.00	5.00

**Table 2 molecules-28-01051-t002:** Percentage reductions and log10 reductions in *S. aureus* and *P. aeruginosa* obtained from the wipe efficacy test for nonwoven fabrics fabricated from nanocomposite raw cotton fibers with various weight percentages.

Weight Percent in a Blend		0%	1%	2%	5%	10%
*S. aureus*	Percent reduction	41.18%	99.59%	99.69%	99.45%	99.56%
	Log_10_ reduction	0.23	2.39	2.51	2.26	2.36
*P. aeruginosa*	Percent reduction	0	99.99%	99.99%	99.99%	99.99%
	Log_10_ reduction	0	4.11	4.11	4.11	4.11

## Data Availability

The data presented in this study are available wholly within the manuscript.

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
