# Peer review of "Washable Antimicrobial Wipes Fabricated from a Blend of Nanocomposite Raw Cotton Fiber"

_molecules, 2023, doi:10.3390/molecules28031051_

Round 1

Reviewer 1 Report

The manuscript covers the topic that could be interesting to readers of Molecules. It is well written and it offers sustainable approach to synthesis of silver nanoparticles in cotton wipes. It can be accepted for publication in Molecules, but before that the authors should make clear what the methyl methacrylate, butyl methacrylate, and methyl ethyl ketone were used for.

Author Response

Dear Reviewer 1:

On behalf of all co-authors, I am submitting a revised manuscript entitled “Washable Antimicrobial Wipes Fabricated from A Blend of Nanocomposite Raw Cotton Fiber” (molecules-2144524).

We thoroughly considered the reviewers’ comments. Following the reviewers’ suggestions, we revised our manuscript. Changes were indicated by using the "Track Changes" function. We are attaching our point-by-point response to your comments below.

We want to take this opportunity to express our sincere thanks to you for your time and efforts to improve our manuscript.

Comments of Reviewer

The manuscript covers the topic that could be interesting to readers of Molecules. It is well written and it offers sustainable approach to synthesis of silver nanoparticles in cotton wipes. It can be accepted for publication in Molecules, but before that the authors should make clear what the methyl methacrylate, butyl methacrylate, and methyl ethyl ketone were used for.

Response: We used methyl methacrylate, butyl methacrylate, and methyl ethyl ketone to prepare cross-sectioned samples for transmission electron microscopic images. We used a mixture of methyl methacrylate, butyl methacrylate to embed fibers and polymerized it using a UV-cross-linker. We used methyl ethyl ketone to remove this polymer medium after placing the slices of the fibers on a carbon-film-coated copper grid. We inserted these preparation details in the section Materials and Methods.

Reviewer 2 Report

This manuscript was written by Sunghyun Nam et al. and reported that a simple and efficient method was developed to produce washable antibacterial wipes. From my perspective, this manuscript contains information that can interest the scientific community, and I recommend its publication. However, amendments must be made before the final publication. Below are listed my observations.

1.        Please provides the HR-TEM and TEM elemental mapping to analyze the Ag nanoparticles in the revised manuscript or supporting information.

2.        Provide the literature survey about washable antibacterial wipes. And how your washable antibacterial wipes are more efficient than other reported ones.

3.        The authors need to re-check this manuscript for spelling and grammatical errors.

4.        How economical when it comes to production on an industrial level. Did the authors have any estimation?

5.        Why are silver ions reduced to silver nanoparticles?

Author Response

Dear Reviewer 2:

On behalf of all co-authors, I am submitting a revised manuscript entitled “Washable Antimicrobial Wipes Fabricated from A Blend of Nanocomposite Raw Cotton Fiber” (molecules-2144524).

We thoroughly considered the reviewers’ comments. Following the reviewers’ suggestions, we revised our manuscript. Changes were indicated by using the "Track Changes" function. We are attaching our point-by-point response to your comments below.

We want to take this opportunity to express our sincere thanks to you for your time and efforts to improve our manuscript.

Comments of Reviewer 2

This manuscript was written by Sunghyun Nam et al. and reported that a simple and efficient method was developed to produce washable antibacterial wipes. From my perspective, this manuscript contains information that can interest the scientific community, and I recommend its publication. However, amendments must be made before the final publication. Below are listed my observations.

1. Please provides the HR-TEM and TEM elemental mapping to analyze the Ag nanoparticles in the revised manuscript or supporting information.

Response: Following the reviewer’s suggestion, we added the results of the HR-TEM and SAED to analyze the silver nanoparticles. The new data were incorporated into Figure 2 and the corresponding discussion was added to the text (lines 116-128).

2. Provide the literature survey about washable antibacterial wipes. And how your washable antibacterial wipes are more efficient than other reported ones.

Response: We conducted the literature survey but we couldn’t find any studies on developing washable antimicrobial wipes. We are the first to report the idea of fabricating washable antimicrobial wipes using cotton nanotechnology.  

3. The authors need to re-check this manuscript for spelling and grammatical errors.

Response: The manuscript was edited by the American Chemical Society (ACS) Authoring Service for proper English language, grammar, punctuation, spelling, and overall style by two highly qualified English-speaking editors. Its certificate and edits (see the second part) were attached.

4. How economical when it comes to production on an industrial level. Did the authors have any estimation?

Response: The approach proposed in this study is economical in production because it eliminates the need for reducing and stabilizing agents, which are typically required for the chemical systhesis of nanoparticles. Here, raw cotton itself acts as both reducing and stabilizing agents. Furthermore, it uses raw (unscured and unbleached) cotton fiber. In the textile industry, cotton fibers are scoured and bleached, and these processes consume large amounts of water, energy, and chemicals. Considering hydrogen peroxide bleaching, it is estimated to save 0.118 kg/kg textiles for chemicals, 8.34 MJ/kg textiles for energy, and 50 L/kg textiles for water. This discussion was inserted in the section of the discussion.

5. Why are silver ions reduced to silver nanoparticles?

Response: To produce silver nanoparticles, silver ions (Ag+) from a silver precursor need to be reduced to silver atoms or metallic silver (Ag0), which nucleate in small clusters that grow into particles. In this study, raw cotton fiber acted as a reducing agent.

Round 2

Reviewer 2 Report

The authors have revised the manuscript according to the comments. Therefore, I recommend that the manuscript could be accepted for publication.